# Membrane mediated mechanical stimuli produces distinct active-like states in the AT1 receptor

Bharat Poudel[1], Rajitha Rajeshwar T[2] & Juan M. Vanegas [1,2,3] ✉

The Angiotensin II Type 1 (AT1) receptor is one of the most widely studied GPCRs within the context of biased signaling. While the AT1 receptor is activated by agonists such as the peptide AngII, it can also be activated by mechanical stimuli such as membrane stretch or shear in the absence of a ligand. Despite the importance of mechanical activation of the AT1 receptor in biological processes such as vasoconstriction, little is known about the structural changes induced by external physical stimuli mediated by the surrounding lipid membrane. Here, we present a systematic simulation study that characterizes the activation of the AT1 receptor under various membrane environments and mechanical stimuli. We show that stability of the active state is highly sensitive to membrane thickness and tension. Structural comparison of membrane-mediated vs. agonist-induced activation shows that the AT1 receptor has distinct active conformations. This is supported by multi-microsecond free energy calculations that show unique landscapes for the inactive and various active states. Our modeling results provide structural insights into the mechanical activation of the AT1 receptor and how it may produce different functional outcomes within the framework of biased agonism.

G-protein coupled receptors (GPCRs) have about 800 distinguished members[1] and are characterized by seven transmembrane (TM) helices linked together by extracellular and intracellular loops. During activation, the ligand interacts with the extracellular loops of the receptor to reach the binding site, which induces rearrangement of the TM helices and intracellular loops to facilitate interactions with its G protein partners. GPCRs play an essential biological role in numerous signaling processes including touch and pain sensation, growth and development, regulating vision, and hormone response among many others[2–4]. A large number of GPCRs are associated with conditions such as depression and cardiovascular regulation, where a significant number of prescribed drugs directly interact with proteins such as the serotonin and angiotensin II (AngII) receptors[5]. The AngII type 1 (AT1) receptor is a member of the GPCR type A family whose function is

directly linked to diseases such as hypertension and congestive heart failure[6–8]. The natural agonist for the AT1 receptor is the vasoconstricting peptide AngII. However, the AT1 receptor can also be mechanically activated by membrane stretch in the absence of AngII[9–12]. While mechanical activation of the AT1 receptor has not been as widely studied compared to agonist binding, it has been recently reported that mechanical activation of AT1R contributes to abdominal aortic aneurysm formation[13].

Activation of the AT1 receptor results in coupling of heterotrimeric G proteins and subsequent recruitment of β-arrestins. Stimulation by the balanced agonist AngII triggers signaling through both Gαq and β-arrestin[14]. However, the AT1 receptor may also be activated by biased agonists that prefer a particular effector[14]. Biased agonists such as the peptides TRV055 and TRV056 highly favor Gαq

[1]Materials Science Graduate Program, The University of Vermont, Burlington, VT 05405, USA. [2]Department of Physics, The University of Vermont, Burlington, VT 05405, USA. [3]Present address: Department of Biochemistry and Biophysics, Oregon State University, Corvallis, OR 97330, USA. ✉e-mail: vanegasj@oregonstate.edu

recruitment, while other partial agonists such as S1I8 (sarcosine1,iso-leucine8-AngII) favor β-arrestin, and yet others such as TRV023 and TRV026 only recruit β-arrest without a Gα partner[11,14–18]. Furthermore, mechanical activation via membrane stretch transduces signaling through both β-arrestin and Gαi[10,11,14,15,19].

Structures of the AT1 receptor in both the active and inactive conformations have been recently determined by X-ray crystallography[17,20–22]. The inactive state was stabilized by binding of the inverse agonist olmesartan[21] or the antagonist ZD7155[20], whereas the active conformation was obtained by binding of S1I8 or AngII in addition to a single-domain antibody fragment, a G-protein mimicking nanobody[17,22]. The salient structural differences between the active and inactive conformations of the AT1 receptor include the orientation and positioning of the transmembrane helices and intracellular loops. During activation, transmembrane helix 6 (TM6) undergoes a large outward movement (-11 Å) on the intracellular side that is accompanied by a smaller outward movement of TM5 (-4 Å) on the same side[22]. Additional intracellular changes in the active structure include the reorganization of intracellular loop 2 (ICL2) to form a short α-helix as well as repositioning of the short amphipathic helix 8 (H8). The highly conserved NPxxY motif also becomes re-oriented in the active configuration which enables hydrogen bonding between tyrosines Y302[7.53] and Y215[5.58] [22]. Superscripts follow the Ballesteros-Weinstein numbering scheme, which is based on the presence of highly conserved residues in each of the seven TM helices. On the extracellular side TM4 undergoes a moderate inward movement (-4 Å). The structural changes observed in the crystallographic studies have been confirmed by double electron-electron resonance (DEER) experiments, which also demonstrated that the receptor structure is highly dynamic[16].

It was originally proposed that mechanical stretch induced the secretion of AngII which would sequentially bind the AT1 receptor and induce activation[23]. However, Zou et al. later demonstrated that the AT1 receptor can be activated by mechanical stretch in cardiomyocytes in the absence of AngII and can cause cardiac hypertrophy in vivo[9]. This was confirmed by patch-clamp experiments showing that the AT1 receptor can be directly activated by membrane stretch without AngII[24,25]. Cysteine accessibility scanning also showed that TM7 of the AT1 receptor undergoes an anticlockwise rotation and a shift in response to membrane mechanical stretch[26]. Furthermore, a recent study showed that mutations in the amphipathic H8 can significantly reduce the mechanical response of the AT1 receptor while the effects of AngII remain unchanged[27]. In addition to membrane stretch, membrane shear may also activate the AT1 receptor[28]. The AT1 receptor is a member of a growing list of mechanosensitive GPCRs that includes GPR68[29], another critical signaling component in cardiovascular pathophysiology, the bradykinin receptor B2 (BDKRB2)[30], and the dopamine receptor D5R[31] among many others. The latter two receptors play an important role in sensing shear stress and hypotonicity in endothelial cells[12]. Despite the significant importance of mechanical activation in GPCRs such as the AT1 receptor, very little is known about the structural changes induced on these proteins by external physical stimuli mediated by the surrounding lipid membrane.

Here, we present a systematic molecular dynamics (MD) simulation study characterizing the activity of the AT1 receptor in different membrane environments and under tension to understand how mechanical stimuli modulate its function. We simulated the apo receptor in membranes with varying acyl chain lengths and lipid headgroup chemistries to probe the effects of membrane thickness and spontaneous curvature on activation. Through comparison with X-ray structures of the AT1 receptor in its active and inactive states as well as simulations with bound agonists, we show that the stability of the receptor's active configuration depends strongly on membrane thickness and spontaneous curvature, which may be tuned by both

membrane composition and/or tension. We show that membrane and ligand-induced activation result in configurations with distinct structural features determined by unique free energy landscapes.

## Results

### Acyl chain length modulates AT1 receptor activation

We first characterize the behavior of the AT1 receptor in single component PC membranes with different chain lengths including DMPC (di-C14:0), POPC (C16:0,C18:1), and SOPC (C18:0,C18:1), which are all fluid at 37 °C. For all 3 membranes, the AT1 receptor was simulated in the apo state, with no ligand or other proteins bound based on initial atomic positions taken from the crystal structure in an active conformation (see "Methods"). Each system was simulated for 25 ns with the protein under position restraints to allow for equilibration of the surrounding membrane, followed by 2 μs unrestrained simulations run in duplicate with different initial velocities (see "Methods"). We first focus on the distance between transmembrane (TM) helices 1 and 6, one of the key structural features considered during activation. The time evolution of the TM1-TM6 distance for one of the replicas (Fig. 1a) shows a strong dependence on bilayer thickness where the distance for the receptor embedded in SOPC (the longest chain tested) rapidly decreases by more than 7 Å as it approaches the value observed in the crystal structure of the inactivated AT1 receptor bound to a selective antagonist[20] (PDBID: 4YAY). Supplementary Fig. 1 shows the time traces for the second replica. The TM1-TM6 distance remains stable near the active state value, as measured in the crystal structure with AngII-bound (PDBID: 6OS0), for the intermediate chain POPC membrane and has a moderate decrease (2–3 Å) for the shortest chain DMPC system (Fig. 1a). We further validated this in simulations using the CHARMM36 forcefield (see "Methods") were we observe the TM1-TM6 distance stable in POPC while it decreases over time in the SOPC membrane (Supplementary Fig. 2a). A similar pattern is observed for the TM1-ICL2 distance (Fig. 1b), where these two protein elements become closer to each other (inactive) when the receptor is in SOPC, while they remain further apart (active) in the thinner POPC and DMPC membranes. However, other structural features including the distances between TM5 and ICL2 (Fig. 1c), TM6 and H8 (Fig. 1d), and TM3 and TM6 (Fig. 1e) show that the active state is favored in the POPC membrane, while in both DMPC and SOPC the distances approach the inactive state values. The distance between the Y302[7.53] hydroxyl, located in the NPxxY motif, and the Y215[5.58] hydroxyl in TM5 shows frequent fluctuations between the values observed in the active (-5 Å) and inactive configurations (-12 Å) in all three PC membranes (Supplementary Fig. 3). This suggests that membrane thickness may not have a strong influence on the orientation of the NPxxY motif.

In addition to relative distances, we also characterize the propensity of ICL2 to be in an α-helical configuration as this short loop forms a helix in the active state, while it is an unstructured coil in the inactive conformation. The helicity of ICL2 is likely to be important for AT1R activation as it significantly changes the orientation of residues R137 and R140 in the loop and how they interact with the receptor's highly conserved DRY motif (D125[3.49]/R126[3.50]/Y127[3.51]). This may also have an effect on the binding of other partners such as G-proteins. The α-helicity of ICL2 for the apo AT1 receptor (Fig. 1f) decreases in all three membranes tested, but appears most stable in POPC. It is completely lost in both replicas by the end of the simulations in the DMPC and SOPC membranes, while in POPC it is partially retained in the first replica and fully stable in the second replica. Together, these results indicate that there is an optimal membrane thickness that favors an active-like conformation, near that of POPC, while thicker or thinner membranes promote inactivation.

### Tension and PE stabilize the active state

Having established that the AT1 receptor is sensitive to changes in bilayer thickness, we further characterize other variations in the local

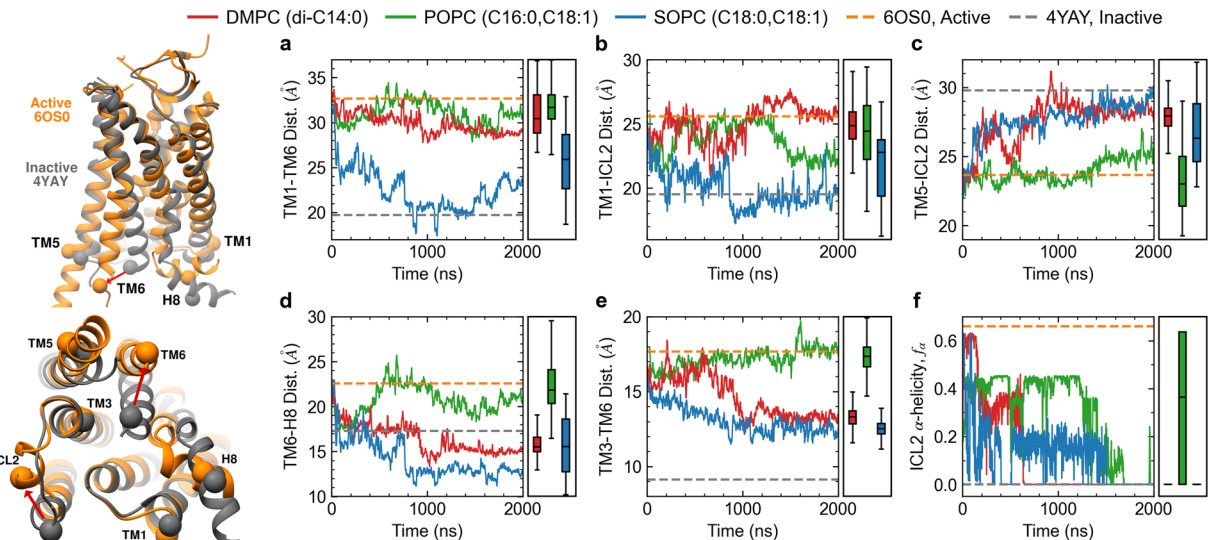

**Fig. 1 | Time evolution of apo AT1 receptor simulations in PC membranes of varying chain lengths.** Key structural features including the distances between TM1-TM6 (**a**), TM1-ICL2 (**b**), TM5-ICL2 (**c**), TM6-H8 (**d**), and TM3-TM6 (**e**), as well as the α-helicity of ICL2 (**f**) indicate that the active state is stable in the POPC (green lines) membrane, while thicker (SOPC, blue lines) or thinner (DMPC, red lines) membranes promote inactivation. Rectangular boxes on the right of each panel show box and whiskers plots including median, quartiles, and extrema of the combined data from the two replicas of each system over the last 500 ns (n = 2 independent simulations, 200,000 time points analyzed). Dashed gray and orange lines show values from crystal structures of the inactive receptor bound to a selective antagonist (4YAY) and active receptor bound to AngII (6OS0), respectively. Inset on the left shows ribbon representations of these structures highlighting the Cα atoms used to compute distances (see "Methods" for details).

membrane environment including application of membrane tension and addition of PE lipids. We chose to test both of these effects on the SOPC membrane system based on our earlier results that the apo receptor tends to return to the inactivated state spontaneously when embedded in this long-chain lipid.

We first characterize the effect of membrane tension on the stability of the active state in the apo AT1 receptor. To test this, we simulated the receptor in an SOPC membrane under a range of tensions from 5 to 20 mN/m. High tension values are necessary computationally due to finite size effects that limit long-range fluctuations, yet these values likely correspond to much lower experimental tensions (see Marsh[32]). As the far-field tension applied over the simulation box may be slow to equilibrate, e.g., the membrane area may need several tens of nanoseconds to reach a steady state, we restrained the AT1 receptor to the active configuration for the first 100 ns of the simulation to prevent inadvertent deactivation of the receptor while the membrane is tensed. Analysis of intra-protein distances and ICL2 α-helicity over the last 500 ns of the 2 μs simulations (combined over the 2 replicas) shows that membrane tension is effective in stabilizing an active-like state of the AT1 receptor (Fig. 2).

The TM1-TM6 distance remains stable in the range of 28–32 Å across all tensions tested (Fig. 2a), which is just below the active state experimental value of 33 Å. Simulation of the apo receptor in SOPC with a tension of 15 mN/m using the CHARMM36 forcefield also shows the TM1-TM6 distance remain stable near the active value (Supplementary Fig. 2b). The TM1-ICL2 distance (Fig. 2a) also remains high, near the active state value, for all tensions tested except at 15 mN/m where it decreases moderately. In contrast to this, the TM5-ICL2 (Fig. 2c) distance shows a different pattern where low (5 mN/m) and high (20 mN/m) tensions result in those two protein elements returning to their inactive-like relative positioning, while intermediate tensions sustain the closer active-like configuration. Similarly, the TM6-H8 (Fig. 2d) distance remains closest to the active state values at tensions of 10 and 15 mN/m. The TM3-TM6 distance (Fig. 2e) remains near the active values for all tensions tested. The α-helicity of ICL2 is completely lost for the low (5 mN/m) and high (20 mN/m) tensions, but it is partially stabilized for intermediate tensions (Fig. 2f). Although the

Y302[7.53]-Y215[5.58] distance has frequent fluctuations in SOPC without tension, low (5 mN/m) and intermediate (10 mN/m) tension values seem to stabilize the orientation of the NPxxY motif (Supplementary Fig. 3). These tension results are consistent with the data presented in the previous section, where we observed that the active state of the AT1 receptor was stabilized by the intermediate acyl chain POPC membrane. As lipid bilayers are largely incompressible, membrane tension induces a change in thickness inversely proportional to the area expansion as can be observed in the peak-to-peak distance of the bilayer mass density (Fig. 3). It is notable that the values of the bilayer thickness for SOPC at tensions of 10 and 15 mN/m are near the value observed for POPC since these intermediate tensions are most effective in stabilizing an active-like state of the AT1 receptor. Application of tension onto POPC membranes produces similar outcomes to those observed in SOPC under tension (Supplementary Fig. 4). However, the optimal tension for stability of the active-like state in POPC is lower (5 mN/m) compared to SOPC. This result is consistent with the observation that the bilayer thickness of POPC with a tension of 5 mN/m is in-between the values for SOPC with 10 and 15 mN/m (Supplementary Fig. 5).

While changes in bilayer thickness, tuned by acyl chain length or membrane tension, clearly modulate the activity of the AT1 receptor in our simulations, there may be other mechanical stimuli that could also play a role in activation. PE lipids are known to shift the equilibrium between states in metarhodopsin[33–37], which may be due to direct lipid-protein interactions or changes in the mechanical state of the membrane as the PE lipids have a more negative spontaneous curvature due to their smaller headgroup compared to PC. We test the effect of PE lipids on the activity of the AT1 receptor by embedding it in a SOPC:SOPE (1:1) membrane and characterizing its structural properties over time as shown for one of the replicas in Fig. 4. Supplementary Fig. 6 shows the time traces for the second replica. Analysis of intra-protein distances and ICL2 α-helicity shows that addition of PE to the membrane is effective in stabilizing an active-like state of the AT1 receptor (Fig. 4). The TM1-TM6 distance (Fig. 4a) remains stable near 29-30 Å for the duration of the simulation in the SOPC:SOPE membrane. A similar pattern is observed in the TM1-ICL2 and TM5-ICL2 distances (Fig. 4

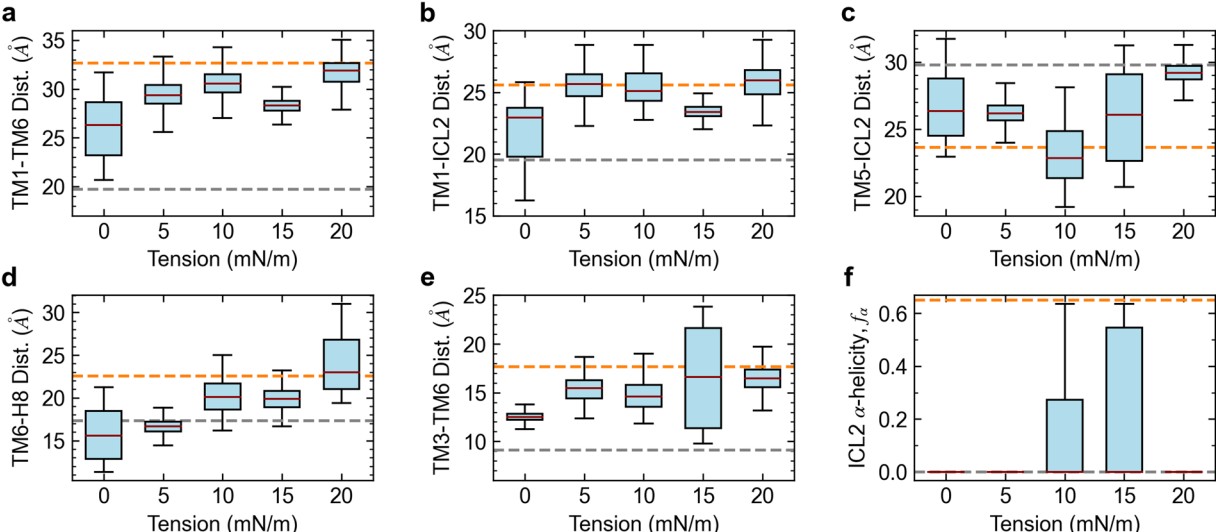

**Fig. 2 | Stability of the AT1 receptor in SOPC membranes under tension (5–20 mN/m) starting from an active configuration.** Panels **a**–**f** show the average intra-protein distances between TM1-TM6, TM1-ICL2, TM5-ICL2, TM6-H8, and TM3-TM6 as well as the α-helicity of ICL2, respectively. Values shown as box and whiskers plots including median, quartiles, and extrema of the combined data from the two replicas over the last 500 ns of the 2 μs simulations ($n = 2$ independent simulations, 200,000 time points analyzed). Dashed gray and orange lines show values from crystal structures of the inactive receptor bound to a selective antagonist (4YAY) and active receptor bound to AngII (6OS0), respectively. Tensions of 10–15 mN/m are most effective in stabilizing an active-like AT1 receptor configuration.

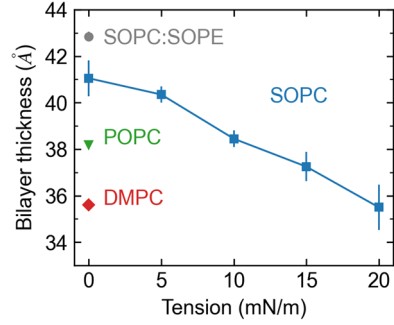

**Fig. 3 | Bilayer thickness for different membrane systems.** The SOPC bilayer thickness under a tension of 10 mN/m closely matches the value observed for the POPC membrane without tension. Bilayer thickness was computed by measuring the peak-to-peak distance of the average bilayer mass density over the last 500 ns of the two replicas ($n = 2$ independent simulations, 200,000 time points analyzed). Error bars show the standard deviation from the mean. Data shown for DMPC (red diamond), POPC (green triangle), SOPC with and without tension (blue squares), and SOPC:SOPE (gray circle).

panels b and c) where the values remain close to those of the active structure for SOPC:SOPE, while they transition to the inactive state values in the pure PC membrane. The TM6-H8 distance (Fig. 4d) and TM3-TM6 (Fig. 4e) distances take values in-between the active and inactive states, while the ICL2 α-helicity (Fig. 4e) is gradually lost over the course of the 2 μs simulation. The presence of PE appears to stabilize the orientation of the NPxxY motif as the $Y302^{7.53}$-$Y215^{5.58}$ distance has higher fluctuations at the early stages of the simulation, but stabilizes near the active state value after 1 μs (Supplementary Fig. 3).

Stabilization of an active-like configuration of the AT1 receptor in the SOPC:SOPE membrane is remarkable given that the bilayer thickness, $42.9 \pm 0.2$ Å, is higher in this membrane compared to pure SOPC, $41.1 \pm 0.8$ Å. The slight increase in thickness for the PE-containing membrane is due to the smaller area per lipid of PE vs PC. While the effects of the PE lipid cannot be readily separated between specific lipid-protein interactions vs. overall changes to the membrane physical/mechanical properties, our simulations do not indicate significant

differences in the interactions between the receptor and either the PC or PE lipids. The observed number of protein H-bonds with PC lipids is $42 \pm 3$ while with PE lipids it is $49 \pm 5$, averaged over the last 200 ns of the two replicas. While these two values changed during the course of the simulation, their ratio remained nearly the same throughout.

As mentioned in the Introduction, a recent experimental study by Erdogmus et al.[27] showed that sensitivity to membrane stretch in the AT1 receptor can be disrupted by substituting the hydrophobic Phe residues F309 and F313 in H8 with Pro without altering the agonist-induced activation. We explore the effects of this modification by simulating the F309P/F313P double mutant in four different membrane conditions: SOPC, SOPC + 10 mN/m tension, SOPC:SOPE, and POPC. Data for these mutants are summarized in Supplementary Figs. 7–10. In the pure SOPC bilayer without tension, some of the features of the F309P/F313P mutant resemble the wild-type AT1 receptor as the TM1-TM6, TM6-H8, and TM3-TM6 distances partially or completely return toward the inactive state values (Supplementary Figs. 7 and 8). However, other distances including TM1-ICL2 and TM5-ICL2 remain close to the active state values in the double mutant unlike the wild-type. The ICL2 α-helicity in the double mutant also remains stable for a large portion of the 2 μs simulation unlike the WT where the α-helicity is lost after a few hundred nanoseconds.

The distances and α-helicity observed in the double mutant receptor in the POPC, SOPE:SOPE, and SOPC + 10 mN/m membranes (Supplementary Figs. 7 and 8) remain in the vicinity of the active state values with the exception of the TM6-H8 pair. The $Y302^{7.53}$-$Y215^{5.58}$ distance in the NPxxY motif explores both the active and inactive values between the two replicas for the F309P/F313P mutant in the SOPC and SOPC:SOPE membranes, while remaining stably near the active state values in the POPC and SOPC + 10 mN/m tension systems (Supplementary Fig. 9). Characterization of the insertion depth of H8 relative to the cytoplasmic lipid headgroups in the WT and F309P/F313P mutant (Supplementary Fig. 10) show that the less hydrophobic helix in the double mutant remains close to the glycerol phosphate in all four systems tested, while H8 in the WT inserts more deeply into the SOPC and SOPC + 10 mN/m membranes. Although some structural features of the F309P/F313P mutant still respond to bilayer thickness, such as the TM1-TM6 distance, it appears that the insertion of H8 relative to the lipid headgroups plays an important role in the AT1

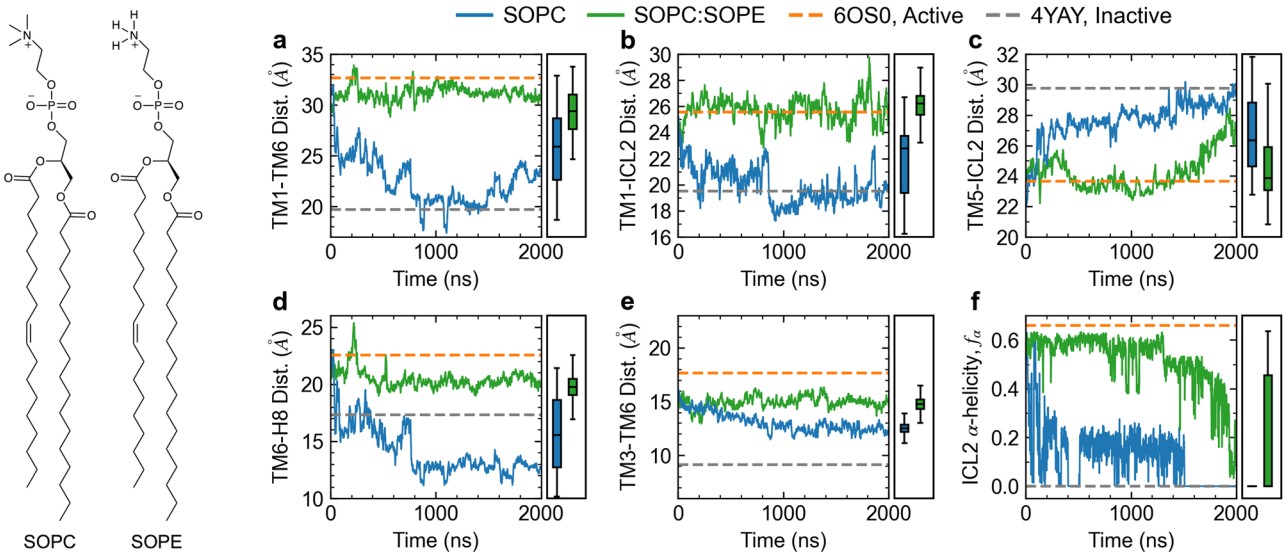

**Fig. 4 | Time evolution of apo AT1 receptor simulations in SOPC and SOPC:SOPE (1:1) membranes.** Key structural features including the distances between TM1-TM6 (**a**), TM1-ICL2 (**b**), TM5-ICL2 (**c**), TM6-H8 (**d**), and TM3-TM6 (**e**), as well as the α-helicity of ICL2 (**f**) indicate that the active state is stable in the SOPC:SOPE (green lines) membrane, while the pure SOPC membrane promotes inactivation. Rectangular boxes on the right of each panel show box and whiskers plots including median, quartiles, and extrema of the combined data from the two replicas of each system over the last 500 ns ($n = 2$ independent simulations, 200,000 time points analyzed). Dashed gray and orange lines show values from crystal structures of the inactive receptor bound to a selective antagonist (4YAY) and active receptor bound to AngII (6OS0), respectively. Inset on the left shows the structure of SOPC and SOPE lipids.

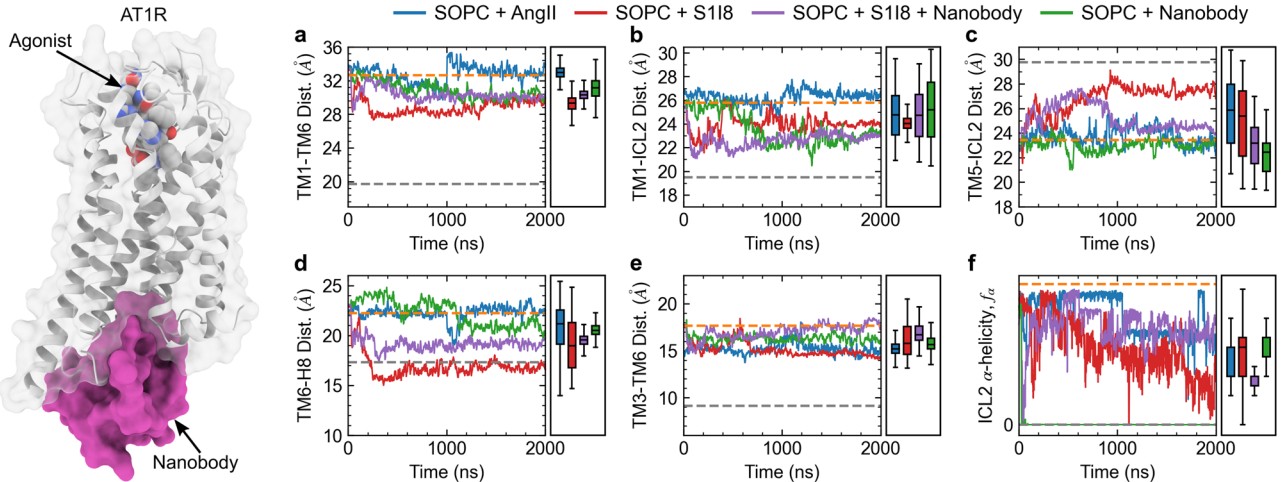

**Fig. 5 | Time evolution of AT1 receptor simulations in SOPC bound to the agonists AngII and S1I8 as well as the AT110i1 nanobody.** Panels **a**–**f** show the AT1 receptor intra-protein distances between TM1-TM6, TM1-ICL2, TM5-ICL2, TM6-H8, and TM3-TM6, as well as the α-helicity of ICL2, respectively. Rectangular boxes on the right of each panel show box and whiskers plots including median, quartiles, and extrema of the combined data from the two replicas of each system over the last 500 ns ($n = 2$ independent simulations, 200,000 time points analyzed). Dashed gray and orange lines show values from crystal structures of the inactive receptor bound to a selective antagonist (4YAY) and active receptor bound to AngII (6OS0), respectively. Only the simulation of AT1 receptor with AngII bound (blue lines) remains stable near the active state configuration, while other combinations of AT1R with S1I8 (red lines), AT1R with S1I8 and nanobody (purple lines), and AT1R with nanobody only (green lines) adopt intermediate conformations.

receptor's sensing of membrane mechanical properties. Short amphipathic helices that sit roughly parallel to the membrane appear to be a common motif in mechanosensitive proteins[38].

## Activation by agonists and nanobody complexes

We now compare the membrane-induced activation with that of the full agonist AngII and partial agonist S1I8 (sarcosine1,isoleucine8-AngII). We also include for completeness simulations with the AT110i nanobody, which is a yeast antibody fragment with high affinity for the AT1 receptor bound to AngII[22] and therefore it stabilizes the active conformation. For these systems, we chose to simulate the AT1 receptor in SOPC following the same protocol as in the previous section, where the

initial protein structure is taken from the active state. Our simulations in SOPC probe the capacity of the agonists and/or nanobody to stabilize the active configuration as was done in the previous section. While the apo AT1 receptor in the SOPC membrane approaches an inactive state after a few hundred nanoseconds (Fig. 1), the AngII bound AT1 receptor appears stable in the active configuration for the duration of the 2 µs simulation as shown for one of the replicas in Fig. 5. Supplementary Fig. 11 shows the time traces for the second replica. When bound to AngII, the TM1-TM6, TM1-ICL2, TM6-H8, and TM3-TM6 distances remain near the active state values, while the TM5-ICL2 distance partially returns to the inactive state value and the ICL2 α-helicity is also partially lost (Fig. 5). Simulation of the AngII-bound receptor in SOPC

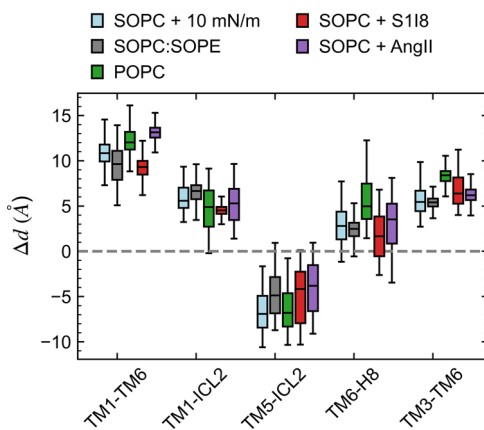

**Fig. 6 | Change in intra-helical distances between active and inactive states,** $\Delta d = d_{\text{active}} - d_{\text{inactive}}$**, for AT1 receptor in SOPC with no agonist bound and 10 mN/m tension (light blue), in SOPC:SOPE (gray), in POPC (green), in SOPC with S1I8 bound (red), and in SOPC with AngII bound (purple).** Values shown as box and whiskers plots including median, quartiles, and extrema of the combined data from the two replicas over the last 500 ns of the 2 μs simulations ($n = 2$ independent simulations, 200,000 time points analyzed). The 4YAY crystal structure of the AT1 receptor with a selective antagonist bound was used as the reference inactive structure.

using the CHARMM36 forcefield also shows the TM1-TM6 distance remain stable near the active value (Supplementary Fig. 1b). When bound to the partial agonist S1I8, the AT1 receptor appears to adopt an intermediate state as the TM1-TM6 distance decreases by 5-6 Å and other distances such as the TM1-ICL2, TM5-ICL2, and TM6-H8 decrease as well (Fig. 5). Additional binding of the AT110i1 nanobody to the AT1 receptor appears to further stabilize the active state as the TM5-ICL2, TM6-H8, and TM3-TM6 distances remain closer the active state values (Fig. 5). The AT110i1 nanobody also plays an important role in stabilizing the orientation of the NPxxY motif as the Y302[7.53]-Y215[5.58] distance remains near the active state value for the two nanobody containing simulations, while it undergoes large fluctuations in the two agonist-only simulations (Supplementary Fig. 3).

**Energetics of distinct membrane and agonist-induced states**

The results shown in the previous sections indicate that there is no single active state for the AT1 receptor, but instead there are multiple active-like conformations induced by different agonists or mechanical stimuli. The change in intra-helical distances between the active and inactive states, $\Delta d = d_{\text{active}} - d_{\text{inactive}}$, of some of the systems studied here illustrates how the AT1 receptor displays distinct structural features beyond dynamical fluctuations that may take place during the microsecond simulations (Fig. 6). We further explore the nature of these distinct active and inactive states through free energy calculations using the locally distributed tension (LDT) collective variable (CV) method[39].

LDT applies a tension-mimicking bias focused on the lipid rim around the receptor protein for rapid and systematic exploration of membrane-mediated configurational changes without having to apply arbitrary forces on protein residues (see "Methods"). We combined the LDT CV with multiple-walker well-tempered metadynamics[40,41] to estimate the free energy of the AT1 receptor under four different membrane conditions without an agonist: SOPC, SOPC with 10 mN/m tension, SOPC:SOPE (1:1), and POPC. We also characterized the free energy of two agonist-bound systems including AngII and S1I8 in SOPC membranes. For each system, we used 8 walkers with initial configurations taken at various intervals from the equilibrium trajectory, and each walker was simulated for >200 ns for a combined simulation time of >1.6 μs (see "Methods"). While the system is biased via the LDT CV, we present our free energy estimations in Fig. 7 as 2-dimensional surfaces based on the TM1-TM6, TM6-H8, TM1-ICL2, and TM5-ICL2

distances using the histogram re-weighting technique[42]. Convergence of the free energy calculations was measured by the root-mean-squared deviation of the overall surfaces as a function of the total simulation time (Supplementary Fig. 12).

Beginning with the AT1 receptor in the inactive state in a pure SOPC membrane (Fig. 7a), the free energy surfaces (FES) show clear minima near the distances observed in the 4YAY inactivated crystal structure with selective antagonist-bound[20] (gray markers). Similarly, the FES for the AngII-bound receptor in SOPC (Fig. 7b) have well-defined minima centered at the distances observed in the 6OS0 crystal structure with AngII bound[17]. In contrast to this, the FES for the S1I8-bound receptor (Fig. 7c) show minima located at intermediate distances distinct from those observed in the AngII-bound FES. The high energy (>7 kcal/mol) in the region marked by the 6OS0 distances when the receptor is bound by the partial agonist S1I8 indicates that the protein is highly unlikely to visit such configuration. The FES for the receptor bound to S1I8 and/or the AngII-specific nanobody (Supplementary Fig. 13) show minima closer to the 6OS0 active state, in agreement with our structural analysis in the previous section (Fig. 5), but still different from the AngII-bound conformation.

In the case of membrane-mediated activation of the apo AT1 receptor, the FES for the SOPC system with applied tension (Fig. 7d) and SOPC:SOPE (Fig. 7e) show stability of the protein in distinct regions that overlap some of the distances observed in the AngII and S1I8 bound configurations. Surprisingly, the minima observed in the FES for the thinner POPC membrane (Fig. 7f) closely match the distances observed in the AngII-bound structure, yet this energy landscape has distinct features compared to the 10 mN/m SOPC system suggesting that the effect of tension goes beyond just changing the bilayer thickness.

**Stability of active states at long timescales**

We further test the stability of the active state of the AT1 receptor in long time-scale simulations using the CHARMM36 FF in the specialized Anton 2 supercomputer (see "Methods"). Earlier modeling studies including multi-microsecond simulations and enhanced sampling techniques have shown that other GPCRs such as the β2-adrenergic and muscarinic receptors may decay from active to inactive states even when bound to agonists[43,44]. Starting from the active state structure, we simulated three systems for 20 μs each including the apo receptor in SOPC, apo receptor in SOPC with a membrane tension of 10 mN/m, and AngII-bound receptor in SOPC. The time courses for key structural distances, including the TM1-TM6 and Y302[7.53]-Y215[5.58] NPxxY motif, of these three simulations are shown in Fig. 8. Focusing first on the apo AT1 receptor in the tensionless SOPC membrane (blue curves in Fig. 8), we observe the protein returning to inactive state values across all distances within a few microseconds consistent with our shorter simulations. It is notable that the TM1-TM6 distance, which undergoes the largest motion, frequently returns to intermediate distances (24−28 Å) after visiting the fully inactive value of ~20 Å (Fig. 8a). This result highlights the dynamic nature of the AT1 receptor in its unbound form.

For the apo receptor under 10 mN/m tension (green curves in Fig. 8), the various structural features appear stable for extended periods of time, although the receptor readily visits intermediate configurations as observed in the TM1-TM6 distance (Fig. 8a) between $t = 4$ and $t = 11$ μs. This suggests that tension does not lock the receptor in an active state, but rather favors activation by increasing the probability of occupying this state. When bound to AngII, the AT1 receptor shows the greatest stability during the course of the 20 μs simulation with all distances remaining close to the active state values except for the TM1-ICL2 and TM5-ICL2 distances (Fig. 8b, c), which show larger variability. The latter two distances depend on the α-helicity of the short ICL2 loop which is lost within the first few microseconds for all three systems, although it is partially recovered intermittently during the course of the simulations (Supplementary Fig. 14). The loss of α-helicity in ICL2 in

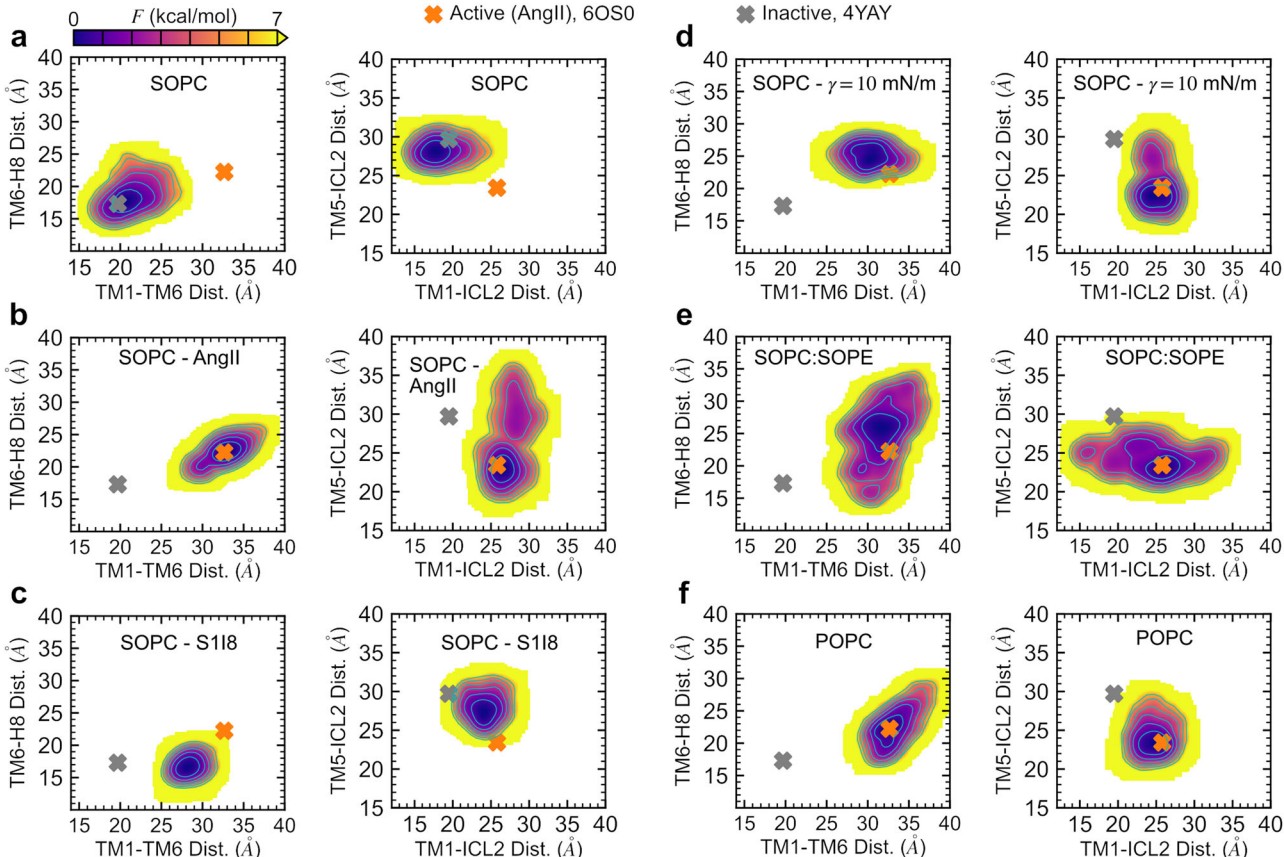

**Fig. 7 | Free energy estimation of AT1 receptor under various conditions. a** apo in SOPC, **b** AngII-bound in SOPC, **c** S1I8-bound in SOPC, **d** apo in SOPC with 10 mN/m tension, **e** apo in SOPC:SOPE, and **f** apo in POPC. Free energy estimated with LDT multiple-walker well-tempered metadynamics and plotted as two-dimensional surfaces through histogram re-weighting using the TM1-TM6, TM6-H8, TM1-ICL2, and TM5-ICL2 distances (see "Methods"). Energy scale shown on top left with lowest values shown in dark blue and highest values shown in yellow (≥7 kcal/mol). Isocontour lines shown in cyan color drawn in 1 kcal/mol intervals. Gray markers show distance values from the inactive state crystal structure with receptor bound to a selective antagonist (PDBID: 4YAY) and orange markers show distance values from the active state crystal structure with bound AngII (PDBID: 6OS0).

these long simulations is likely driven by the modified TIP3P water model, which has additional Van der Waals interactions on the H atoms and is the recommended choice for the CHARMM36 membrane FF. This water model is known to destabilize alpha-helical peptides compared to the unmodified TIP3P model as well as experiments[45]. The orientation of the NPxxY motif remains close to the active state values after 20 μs in both the apo receptor under tension and when bound to AngII (Fig. 8f). The structural similarities between the tension-stabilized active state simulation at $t$ = 20 μs and the AngII-bound X-ray crystal structural are shown in Fig. 9a, where the closely matching orientation of the TM and H8 helices can be better observed. A similar comparison of the simulated inactive state, at $t$ = 5 μs in the tensionless SOPC membrane, versus the crystal structure of the receptor bound to an antagonist is shown in Fig. 9b.

## Discussion

We have presented a systematic structural characterization of membrane vs. ligand-induced activation in the AT1 receptor through multi-microsecond atomistic MD simulations and free energy calculations. Our simulations of the apo AT1 receptor in various membranes clearly show that the local lipid environment can modulate protein activation in the absence of an agonist. The apo receptor readily transitions to an inactivated state when simulated in the long-chain SOPC membrane, while it remains in a stable active conformation in the tensioned-SOPC bilayers. This was observed in both short 2 μs simulations with the GROMOS FF as well as long 20 μs Anton 2 simulations with the CHARMM36 FF. The combined results from simulations in membranes with different acyl chain lengths and increasing tensions indicate that there is an optimal bilayer thickness that promotes activation, while thinner or thicker membranes induce inactivation. Furthermore, inclusion of SOPE lipids with SOPC also favored the active AT1 receptor conformation despite the large membrane thickness (greater than the pure SOPC system). This effect could be due to direct lipid interactions with the PE headgroups and/or changes in the internal stress state of the membrane as the PE lipids have a more negative spontaneous curvature. Differences in the size and headgroup chemistries in PC vs. PE lipids will lead to changes in the distribution of attractive and repulsive lateral pressures near the headgroup and hydrophobic interface regions that balance the spontaneous curvature frustration[46–49]. The choice of SOPC, SOPE, and POPC lipids for the majority of the simulations in our study may not reflect the particular composition of any given tissue, yet lipids with 16 and 18 carbon chains as well as PC and PE headgroups are commonly found in mammalian plasma membranes[50,51]. Real biological membranes are far more complex including not only many other headgroup chemistries (e.g., phosphatidylserine and phosphatidylinositol), but also different levels of tail unsaturations, cholesterol, and leaflet asymmetry[49–51]. Spatial variations in lipid composition within the plasma membrane can result in localized changes in the thickness, fluidity, and curvature[48,49]. These can in turn drive both the lateral sorting of proteins to certain regions of the membrane[49,52,53] and also modulate their function[48,49]. For example, the activity of transporters and ATPases has been shown to be strongly dependent on bilayer thickness[48].

Our observations that changes in bilayer thickness and addition of PE lipids modulate activation in the AT1 receptor are in agreement with

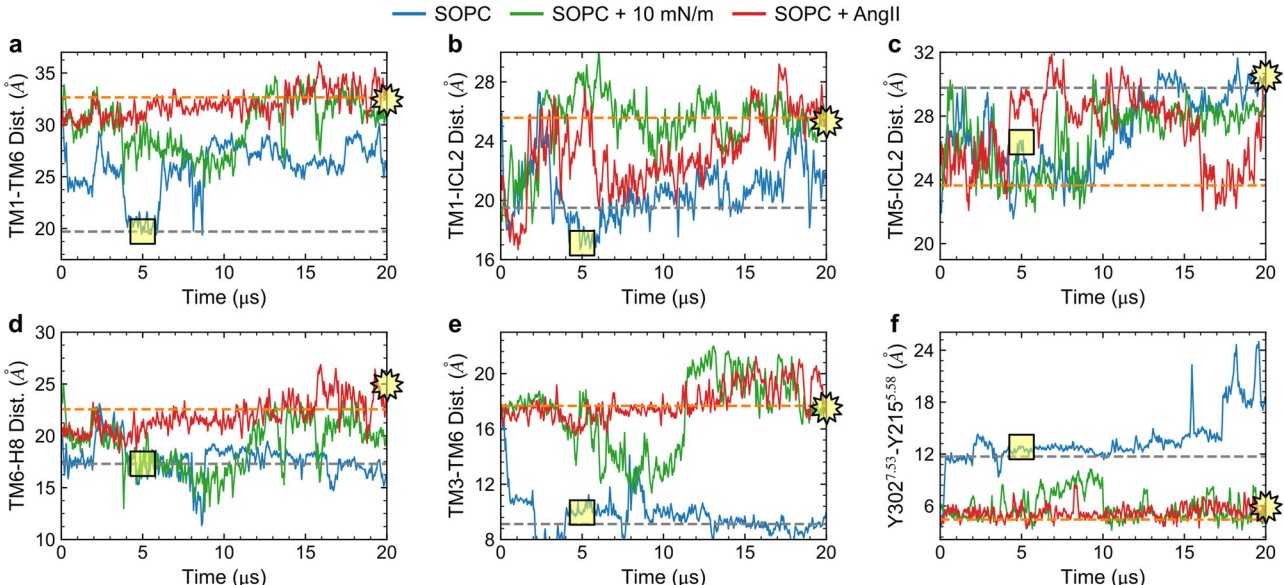

**Fig. 8 | Long time-scale (20 μs) evolution of select AT1 receptor systems simulated with the CHARMM36 FF on the Anton 2 supercomputer.** Data for apo receptor in a tensionless SOPC shown in blue, in SOPC with 10 mN/m tension shown in green, and in SOPC with AngII bound shown in red. Panels **a**–**f** show the intra-protein distances between TM1-TM6, TM1-ICL2, TM5-ICL2, TM6-H8, TM3-TM6, and Y302$^{7.53}$-Y215$^{5.58}$ (NPxxY motif), respectively. Dashed gray and orange lines show values from crystal structures of the inactive receptor bound to a selective antagonist (4YAY) and active receptor bound to AngII (6OS0), respectively. Square and star yellow symbols highlight the values of the distances for the inactive apo system at $t = 5$ μs and active AngII bound system at $t = 20$ μs that are shown in Fig. 9.

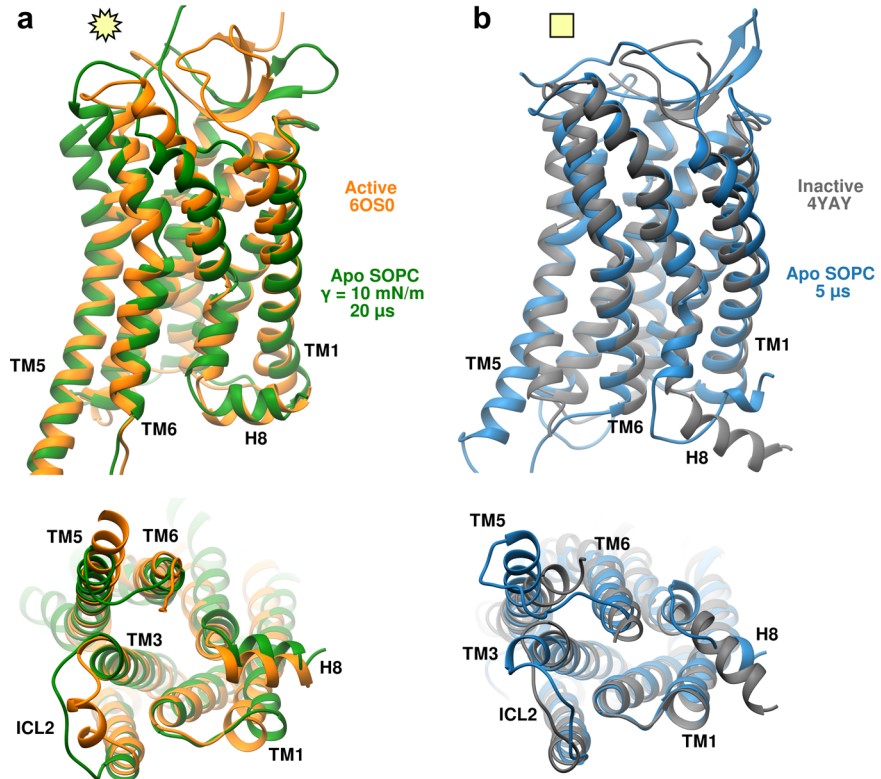

**Fig. 9 | Ribbon representations of long time-scale simulations of the apo AT1 receptor in SOPC membranes using the CHARMM36 FF on the Anton 2 supercomputer. a** Structure after 20 μs with a tension of 10 mN/m compared to the active state crystal structure (PDBID: 6OS0, with AngII-bound). **b** Structure after 5 μs with no tension compared to inactive state crystal structure (PDBID: 4YAY, with antagonist-bound). Structures superimposed by minimizing the RMSD distance.

the well-characterized GPCR system of rhodopsin[48]. Experiments have shown that rhodopsin is exquisitely sensitive to membrane properties such as thickness, hydrophobic mismatch, spontaneous curvature, and lipid headgroup and acyl chain chemistry[37,54,55]. The equilibrium between the metarhodopsin I (MI, inactive) and metarhodopsin II (MII, active) states can be modulated through both direct lipid-protein interactions as well as indirectly through changes in the internal stress state of the surrounding membrane. Lipids that generate negative curvature such as PE shift the MI-MII equilibrium toward the MII state[33–36]. Variations in lipid acyl chain length, ranging from 14 to 24 carbon atoms per chain, also shift the MI-MII equilibrium and suggest that there may be an optimal chain length that favors the MII state[56,57]. Direct interactions with PE lipids and docohexanoic acid (DHA) favor the MII state[57], while addition of cholesterol to PC membranes shifts the equilibrium toward the MI state[58–60].

Our structural analysis based on the distances between critical structural elements of the AT1 receptor (TM1-TM6, TM1-ICL2, TM5-ICL2, TM3-TM6, Y302$^{7.53}$-Y215$^{5.58}$, and TM6-H8) shows that membrane tension and PE lipids produce similar active-like states (Fig. 6). These conformations resemble the AngII and S1I8-induced active states, yet they appear to have distinct structural features. The free energy landscapes of the active states induced by tension, PE, AngII, and S1I8 (Fig. 7) show unique features in each case despite the large overlap in some areas. Recent experimental studies including X-ray crystallography and DEER data have demonstrated that binding of different agonists and nanobodies produces distinct active states in the AT1 receptor[16,17,22]. While the simulations and analysis presented here are limited to the AT1 receptor without its G-protein or β-arrestin partners, our results are well-aligned with the emerging framework of biased-agonism where distinct receptor states interact with different effectors to produce various functional outcomes[14].

## Methods
### GROMOS FF simulations
Molecular dynamics (MD) simulations were carried out with the GROMACS simulation package[61] v2019.5 using the GROMOS 43A1-S3 force-field for lipids and GROMOS 54A7 forcefield for proteins[62,63]. Water was simulated with the SPC/E model[64]. The starting AT1 receptor active state configuration was obtained from Wingler et al.[22] (PDB ID: 6DO1). D74$^{2.50}$ and D125$^{3.49}$ were simulated in their protonated states to maintain the local interactions in the buried low-hydration environment[18]. The loop that connects TM5 and TM6 has been modeled based on the information provided in the crystal structure using the Modeller v9.12 package[65]. Because of this additional loop, the AT1 receptor sequence used is shifted by five residues from TM6 onward compared to other conserved GPCRs residues. For all simulations, the AT1 receptor was embedded into equilibrated membrane patches with 298 lipids (149 in each leaflet) including di-myristoyl-sn-glycero-3-phosphocholine (DMPC), 1-palmitoyl-2-oleoyl-glycero-3-phosphocholine (POPC), 1-stearoyl-2-oleoyl-sn-glycero-3-phosphocholine (SOPC), and a 1:1 mixture of SOPC and 1-stearoyl-2-oleoyl-sn-glycero-3-phosphoethanolamine (SOPE). A sufficient water layer was added to prevent the receptor from interacting with its periodic image (>30,000 water molecules). Chloride ions were added to neutralize the system. Following energy minimization, all membrane-protein systems were equilibrated first for 25 ns with protein-only position restraints ($k$ = 1000 kJ/mol/nm$^2$) under constant temperature (37 °C) and pressure (1 atm.) using a Berendsen thermostat and a semi-isotropic Berendsen barostat. The equilibration was then continued for another 75 ns without the protein position restraints. After equilibration, a production run of 2 μs was conducted at the same temperature and pressure using a Nose-Hoover thermostat and a semi-isotropic Parrinello-Rahman barostat, while saving positions at 5 ps time intervals. For all systems in this study except for POPC under tension, we simulated two replicas starting from the same initial structure but with different initial velocities taken from a Maxwell distribution. Equations of motion were integrated with a leap-frog algorithm using a 2 fs time step. Van der Waals interactions were computed using a plain potential cutoff with a radius of 1.6 nm, and electrostatic interactions were computed using the particle-mesh Ewald method with a real space cutoff 1.6 nm and a Fourier grid spacing of 0.12 nm.

In addition to varying the lipid acyl chains, we also simulated the AT1 receptor in combination with the full agonist AngII or the partial agonist S1I8 (sarcosine1,isoleucine8-AngII), and the single-domain antibody fragment AT110i1 (nanobody or Nb)[22]. The S1I8 peptide differs from AngII in that Asp1 is replaced with N-methyl Gly and Phe8 is replaced by Ile. Coordinates for the AT1 receptor with S1I8 and nanobody were taken from the 6DO1 structure[22]. Orientation of the AngII agonist was based on that of S1I8.

Various systems including the apo AT1 receptor were simulated under tension via the semi-isotropic Berendsen barostat. Tension in the membrane plane ($x−y$) was set to values in the range of 5 to 20 mN/m, while the normal pressure ($z$) was set to 1 atm. Given that the membrane area may take some time to equilibrate under tension, we allowed each system to run for 25 ns with position restraints on the protein in order for the lipid membrane to reach its equilibrium area. This was followed by a 100 ns run without protein position restraints with the Berendsen thermostat. Production runs were then completed with the Nose-Hoover thermostat and semi-isotropic Berendsen barostat for 2 microseconds. The last 200 ns of the trajectory were used for the analysis. Specific details containing the total number of atoms, lipids, water molecules, ions, and system dimensions are provided in Supplementary Table 1.

### CHARMM36 FF and Anton 2 simulations
Additional simulations of the AT1 receptor embedded in POPC and SOPC using the CHARMM36 forcefield[66–70] were conducted for additional validation. Simulations were setup with the CHARMM-GUI webserver[71,72]. Water was simulated with the modified CHARMM TIP3P model (with additional Van der Waals parameters on the H atoms). A sufficient water layer was added to prevent the receptor from interacting with its periodic image (>1.5 nm). Ions were added to neutralize the system and to maintain an ionic concentration of 150 mM. Following energy minimization, all membrane-protein systems were equilibrated first for 25 ns with protein-only position restraints ($k$ = 1000 kJ/mol/nm$^2$) under constant temperature (37 °C) and pressure (1 atm.) using a Berendsen thermostat and barostat (semi-isotropic). After equilibration, a production run of 1 μs was conducted at the same temperature and pressure using a Nose-Hoover thermostat and a Parrinello-Rahman barostat (semi-isotropic), while saving positions at 5 ps time intervals. Equations of motion were integrated with a leap-frog algorithm using a 2 fs time step. Van der Waals interactions were computed using a force-switching cutoff between 1.0 and 1.2 nm. Electrostatic interactions were computed using the particle-mesh Ewald method with a real space cutoff 1.2 nm and a Fourier grid spacing of 0.12 nm. Specific details containing the total number of atoms, lipids, water molecules, ions, and system dimensions are provided in Supplementary Table 2.

Three selected systems were further simulated using the CHARMM36 FF with the Anton 2 supercomputer. Following the 25 ns equilibration with position restraints on the protein atoms as detailed above, we simulated for 20 μs each the AT1 receptor bound to AngII in SOPC, in its apo state in SOPC without tension, and in its apo state in SOPC with a membrane tension of 10 mN/m. The equations of motion were integrated using the multigrator method with a 2.5 fs time step[73]. Short-range forces were evaluated at every time step, and long-range electrostatics were calculated every three time steps using the Gaussian split Ewald method[74]. Van der Waals interactions were computed using a 1.2 nm cutoff. System pressure was kept constant at 1 atm. using a semi-isotropic Martyna-Tobias-Klein (MTK)[75] barostat. Temperature was kept constant with a Nose-Hoover thermostat at 37 °C.

## Free energy calculations

Free energy calculations were obtained through biasing of the membrane with the locally distributed tension (LDT) collective variable (CV)[39]. The LDT CV is defined by a smooth hyperbolic tangent stepping function

$$\xi = \sum_{i=1} \tanh\left(\frac{d_i - d_{\min}}{a}\right), \tag{1}$$

where $d_i$ is the lateral distance from the center of mass of each lipid to the center of mass of the protein, $d_{\min}$ is the minimum distance from the center of mass of the protein, and the constant $a = 1$ determines how rapidly the hyperbolic function reaches unity. We combined the LDT CV with multiple walker[41] well-tempered metadynamics[40] simulations to explore the free energy landscapes in the vicinity of active/inactive states for various systems. All LDT free energy calculations were performed using a PLUMED (v. 2.5-2.7)[76,77] patched version of GROMACS 2019. In the metadynamics runs, the simulations are biased with a time ($t$) dependent potential of the form

$$V(\xi, t) = \sum_{t'}^{t' < t} W \exp\left(-\frac{V(\xi, t')}{k_{\mathrm{B}} \Delta T}\right) \exp\left(-\frac{(\xi - \xi(t'))^2}{2\sigma^2}\right), \tag{2}$$

where $W$ and $\sigma$ are the height and width of the added Gaussian hills, respectively. $\Delta T$ is a fictitious maximum increase in temperature that ensures convergence by limiting the extent of the free energy exploration. At long timescales, one can recover the unbiased free energy, $F(\xi)$, from

$$V(\xi, t \to \infty) = -\frac{\Delta T}{T + \Delta T} F(\xi) + C, \tag{3}$$

where $C$ is an immaterial constant. The value of $\Delta T$ is set by the bias factor parameter, $B = \frac{T + \Delta T}{T}$ and the frequency of addition of Gaussian hills is determined by a fixed deposition rate, $\omega$. The same values of $\sigma = 0.05$, $B = 20$, $W = 1.2$ kJ/mol, and $\omega = 10$ ps were used for all free energy calculations. Eight independent walkers simultaneously contributed to the same metadynamics biasing potential to accelerate sampling. Initial structures for each walker were taken in 10 ns intervals from time 930 to 1000 ns of the production runs. All walkers were then simultaneously run for >200 ns/each using well-tempered metadynamics. Therefore, the combined simulation time to obtain each free energy surface was >1.6 μs (8 walkers × 0.2 μs). Convergence of the free energy profiles was monitored by computing the root-mean-squared deviation of the estimated profiles in 2 ns intervals (per walker). Two-dimensional free energy surfaces based on various intra-protein distances were obtained by re-weighting (unbiasing) the metadynamics trajectories assuming a constant bias during the entire course of the simulation[42].

## Data analysis

For characterization of the structural differences between active and inactive states of the AT1 receptor, we measured the transmembrane distances with the same residues chosen in previous electron paramagnetic resonance studies of activation/inactivation[17]. We chose the Cα atoms from the following residues: F55[1.53] for TM1, R126[3.50] for TM3, K220[5.38] for TM5, D236[6.34] for TM6, R311 for H8, and R139 for ICL2.

Bilayer thickness was estimated by computing the density of phosphorus atoms along z-axis on both leaflets and measuring the peak to peak distance[46]. The α-helicity of ICL2 was predicted using the DSSP algorithm[78]. All figures were plotted using the Matplotlib library[79]. Molecular models of the AT1 receptor were created using UCSF Chimera and ChimeraX[80,81].

## Reporting summary

Further information on research design is available in the Nature Portfolio Reporting Summary linked to this article.

## Data availability

Source data are provided under https://github.com/Bharat-github6/Mechanical-activation-of-AT1-receptor. This github repository contains all data generated from the MD simulations as well as the python scripts used to generate the figures in the main text and Supplementary Information. The Github repository also contains the initial and final configurations for all simulated systems as PDB files. The repository has been archived at Zenodo[82]. The repository also includes examples for how to estimate the free energy surfaces using the LDT bias with well-tempered metadynamics. Any other data or simulation trajectories are available from the corresponding author upon reasonable request. The following crystallographic structures were used in this study: 6DO1 [https://doi.org/10.2210/pdb6DO1/pdb], 4YAY [https://doi.org/10.2210/pdb4YAY/pdb], 6OS0 [https://doi.org/10.2210/pdb6OS0/pdb].

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

## Acknowledgements

J.M.V. and B.P. acknowledge the support of the National Science Foundation through Grant No. CHE-1944892. Computations were performed, in part, on the Vermont Advanced Computing Core supported in part by NSF Award No. OAC-1827314. This work used the Extreme Science and Engineering Discovery Environment (XSEDE), which is supported by National Science Foundation grant number ACI-1548562. XSEDE resources were provided at the San Diego Supercomputing Center (SDSC) Expanse system through allocation TG-BIO210110. Anton 2 computer time was provided by the Pittsburgh Supercomputing Center (PSC) through Grant R01GM116961 from the National Institutes of Health. The Anton 2 machine at PSC was generously made available by D. E. Shaw Research. We thank Professor Ed Lyman for helpful comments and suggestions.

## Author contributions

B.P. setup and carried out MD simulations. B.P. and J.M.V. performed data analysis and created figures. R.R.T. provided analysis scripts. J.M.V. conceived the project and designed simulation studies with input from B.P. and R.R.T. B.P., R.R.T., and J.M.V. contributed to writing of paper, reviewed, and approved it in its final form.

## Competing interests

The authors declare no competing interests.
