## [Peer Review file · Nature Communications]

REVIEWER COMMENTS

Reviewer #1 (Remarks to the Author):

Molecular dynamics (MD) simulations have been applied to characterize the activation of the AT1 receptor under various membrane environments and mechanical stimuli. The simulations showed that there is an optimal membrane thickness that favors activation, near that of POPC, while thicker or thinner membranes promote inactivation of the apo receptor. Intermediate tensions are most effective in stabilizing an active state of the AT1 receptor. Free energy calculations showed distinct conformations for the inactive and various active states of the AT1 receptor.

1. Deactivation of even agonist-bound class A GPCRs in POPC lipids upon removal of G protein have been observed in previous longer MD simulations beyond 1000ns using the Anton supercomputer or enhanced sampling simulations within 1000ns. It's unclear whether the conclusions of this study can be affected by the lengths of the simulations, especially for apo AT1 in Fig. 1 and agonist-bound AT1 in Fig. 4.

Moreover, it may help to run multiple independent MD simulation on each system to improve the statistics and obtain more conclusive results.

2. The TM3-TM6 distance and TM7 NPxxY motif movement have been widely used to characterize activation of class A GPCRs. It could help to examine them in MD simulations of the AT1 receptor as well.

BW numbers are recommended for the discussed GPCR residues.

3. For MD simulations with different membrane tensions, would it help to have control simulations on the POPC lipids which have been mostly used in GPCR simulations? Would deactivation of the AT1 receptor occur regardless of the membrane tension of POPC lipids?

4. For the SOPE vs SOPC lipids, are there any negatively charged residues on the receptor surface that interact differently with the lipid head groups?

5. It's unclear whether S118 is a partial or full agonist. How different is it compared with AngII?

6. More details are needed to support the claim that "our results are in excellent with recent experiments [16,17,22]".

Reviewer #2 (Remarks to the Author):

It is now established that certain GPCRs are influenced by mechanical forces, in particular those present in the apical cell membrane of endothelial cells, which is continuously exposed to the fluid shear stress generated by blood flow. The molecular bases for the mechanosensitivity of GPCRs remain unclear to date. Here, the authors investigate how the AT1 receptor structure is modulated by membrane tension, the presence of bound ligands, lipid thickness and composition, using all-atom MD simulations and free energy calculations. Using a residue-residue distances matrix, the authors observe that certain membrane conditions (POPC, PE and tension) stabilize an active conformation in absence of bound ligand, whereas other conditions (lipids with long or short tails) stabilize a non-active conformation. This study would suggest that membrane thickness could change the conformation of the receptor, potentially linking mechanical membrane deformations to GPCR signaling modulation.

Major concerns

The authors used an arbitrary one microsecond time for all MD stimulations. However, many trajectories are clearly not converged at the end of the run, in particular: DMPC and SOPC runs in Fig1, Apo-SOPC run in Fig3, AT1R+S1I8 in Fig4. Hence, I wonder if these observations are sufficiently sampled to provide meaningful conclusions.

None of the MD simulation has been independently replicated (i.e. with different initial atomic velocities), as it is now customary. Given the modest size of the system, independent replicas are technically feasible, although this may take considerable amount of time for the 14 trajectories shown in Fig 1-4.

Other points

The authors might want to consider discussing the role of Helix 8 as mediating mechanical forces to other class-A GPCRs (see Erdogmus et al., Nat comm 2019).

The authors use the AngII-bound 6OS0 PDB structure as a canonical active state, but other structures with different bias agonist profiles exist (6OS1 and 6OS2). I wonder how the active conformation(s) stabilized by PE and tension would compare to those structures.

Reviewer #3 (Remarks to the Author):

The manuscript presented by Poudel et al describes different activation states of the AT1 receptor under different membrane compositions and the addition of membrane tension. The study is solely based on molecular dynamics simulations which uses hallmark structural changes taken from the available crystal structures of the active and inactive state to evaluate the different membrane effects.

Key findings include the fact that membrane thickness is a key player in the activation state of the AT1R, which should serve as explanation for mechano-stimulation of this receptor. This notion would for sure be of high interest if it would be supported by experimental data. But as it is the paper yields more questions than answers. The fact itself that mechanical stress induces distinct conformational changes in AT1R has already been shown through MD simulations by Yasuda et al in 2008 (DOI: 10.1038/sj.embor.7401157), a study that was not cited by the authors. Even though I am sure there is more to describe in this process, the authors should support their models by experimental data. Artificial membrane-like environments could be used for such a study.

A more general question is where the applied settings for the simulation actually play a physiological role. Are the used short and long fatty acids representative for a certain tissue or cell-type? Is it feasible to assume that pure changes in lipid thickness can appear that induce a change distinguishable from other influencing factors such as headgroup charges or protein and cholesterol distribution?

How can membrane tension lead to a thinner membrane when the same fatty acids are present? Is there a change in the arrangement? How do the enormously high-tension ranges of 5 to 20 mN/m that are required for the simulation relate to physiological force ranges?

Overall, there are some interesting aspects of the paper that once supported by experimental data might provide a deeper understanding of the membrane-induced effects on AT1R signaling.

Reviewer 1

Molecular dynamics (MD) simulations have been applied to characterize the activation of the AT1 receptor under various membrane environments and mechanical stimuli. The simulations showed that there is an optimal membrane thickness that favors activation, near that of POPC, while thicker or thinner membranes promote inactivation of the apo receptor. Intermediate tensions are most effective in stabilizing an active state of the AT1 receptor. Free energy calculations showed distinct conformations for the inactive and various active states of the AT1 receptor.

1. Deactivation of even agonist-bound class A GPCRs in POPC lipids upon removal of G protein have been observed in previous longer MD simulations beyond 1000ns using the Anton supercomputer or enhanced sampling simulations within 1000ns.

We would be happy to examine that study if you could please provide a reference.

It's unclear whether the conclusions of this study can be affected by the lengths of the simulations, especially for apo AT1 in Fig. 1 and agonist-bound AT1 in Fig. 4.

Moreover, it may help to run multiple independent MD simulation on each system to improve the statistics and obtain more conclusive results.

We have increased the length of all our simulations in the study to 2 microseconds and have further simulated a second replica of each system with different initial velocities (sampled from a Maxwell distribution). Together, this comes to four-times the amount of data we had in our original submission without including new simulations that were run to address other concerns. In all new figures we show the time evolution of the distances of first replica along with box and whisker plots that including median, quartiles, and extrema of the combined data from the two replicas of each system over last 500 ns. We don't show the time traces of both replicas in the same figure as this would make the data very difficult to follow as the lines have different time dependances for each replica. We have also run additional simulations using the CHARMM36 force-field of a few key systems to further validate the main message of our study. The methods and figures for this CHARMM36 simulations are described in the Supplementary Material (Supplementary Methods and Fig. S1) and discussed in the main text (page 5 lines 295-301, page 7 lines 380-384, page 10 lines 561-565).

2. The TM3-TM6 distance and TM7 NPxxY motif movement have been widely used to characterize activation of class A GPCRs. It could help to examine them in MD simulations of the AT1 receptor as well.

We have included two additional analyses in our work to characterize the distances between TM3 and TM6, as well as the movement of the TM7 NPxxY motif. For the TM3-TM6 distance we have used residues R125^{3,50} and D236^{6,34}, while for the NPxxY orientation we have chosen the distance between the hydroxyls of Y302^{7,53} and Y215^{5,58}. The new TM3-TM6 distances are included as additional panels in the appropriate figures. The NPxxY motif distance is shown in figures S2 and S6 in the Supplementary Material. We discuss the observations from these two analyses in various portions of the text: Page 2 lines 70-74, page 5 lines 312-320, page 7 lines 401-405, page 8 lines 458-463, page 9 lines 512-518, page 10 lines 574-580.

BW numbers are recommended for the discussed GPCR residues.

We have incorporated the BW numbering scheme for the TM residues (page 2 lines 74-77, page 4 lines 252-254).

3. For MD simulations with different membrane tensions, would it help to have control simulations on the POPC lipids which have been mostly used in GPCR simulations? Would deactivation of the AT1 receptor occur regardless of the membrane tension of POPC lipids?

We ran additional simulations of POPC lipids with 5 mN/m and 10 mN/m tension. Analysis of these systems (Figures S3 and S4 in the Supplementary Material) shows a similar pattern as that observed with the SOPC membranes where mild tensions show increased stability of the active state. However, as POPC is already a thinner membrane, the 'optimal' tension appears to be in the regime of 5-10 mN/m. See page 7 lines 419-427 in the text.

4. For the SOPE vs SOPC lipids, are there any negatively charged residues on the receptor surface that interact differently with the lipid head groups?

The vast majority of H-bonds/charged interactions between protein residues and nearby lipids take place with the phosphate and carbonyl oxygens. We observe few H-bonds between the NH3 of PE and protein residues.

5. It's unclear whether S118 is a partial or full agonist. How different is it compared with AngII?

S118 is considered a partial agonist whereas AngII is the full/balanced agonist. The difference between the two is that S118 has an N-methyl glycine instead of Asp at position 1, and Ile at position 8 instead of Phe. We have mentioned that S118 is a partial agonist in multiple locations in the text to clarify this point (page 3 line 195-196, page 10 line 538-539 and 565-566)

6. More details are needed to support the claim that "our results are in excellent with recent experiments [16,17,22]".

We have removed this sentence from the text in order to avoid ambiguity and we have included a new sentence that better describes the experimental observations that we were referring to (page 13 lines 750-753):

“Recent experimental studies including X-ray crystallography and DEER data have demonstrated that binding of different agonists and nanobodies produces distinct active states in the AT1 receptor [16, 17, 22].”

Reviewer 2

It is now established that certain GPCRs are influenced by mechanical forces, in particular those present in the apical cell membrane of endothelial cells, which is continuously exposed to the fluid shear stress generated by blood flow. The molecular bases for the mechanosensitivity of GPCRs remain unclear to date. Here, the authors investigate how the AT1 receptor structure is modulated by membrane tension, the presence of bound ligands, lipid thickness and composition, using all-atom MD simulations and free energy calculations. Using a residue-residue distances matrix, the authors observe that certain membrane conditions (POPC, PE and tension) stabilize an active conformation in absence of bound ligand, whereas other conditions (lipids with long or short tails) stabilize a non-active conformation. This study would suggest that membrane thickness could change the conformation of the receptor, potentially linking mechanical membrane deformations to GPCR signaling modulation.

Major concerns

The authors used an arbitrary one microsecond time for all MD stimulations. However, many trajectories are clearly not converged at the end of the run, in particular: DMPC and SOPC runs in Fig1, Apo-SOPC run in Fig3, AT1R+S1I8 in Fig4. Hence, I wonder if these observations are sufficiently sampled to provide meaningful conclusions.

None of the MD simulation has been independently replicated (i.e. with different initial atomic velocities), as it is now customary. Given the modest size of the system, independent replicas are technically feasible, although this may take considerable amount of time for the 14 trajectories shown in Fig 1-4.

We have increased the length of all our simulations in the study to 2 microseconds and have further simulated a second replica of each system with different initial velocities (sampled from a Maxwell distribution). Together, this comes to four-times the amount of data we had in our original submission without including new simulations that were run to address other concerns. In all new figures we show the time evolution of the distances of first replica along with box and whisker plots that including median, quartiles, and extrema of the combined data from the two replicas of each system over last 500 ns. We don't show the time traces of both replicas in the same figure as this would make the data very difficult to follow as the lines have different time dependances for each replica. We have also run additional simulations using the CHARMM36 force-field of a few key systems to further validate the main message of our study. The methods and figures for this CHARMM36 simulations are described in the Supplementary Material (Supplementary Methods and Fig. S1) and discussed in the main text (page 5 lines 295-301, page 7 lines 380-384, page 10 lines 561-565).

Other points

The authors might want to consider discussing the role of Helix 8 as mediating mechanical forces to other class-A GPCRs (see Erdogmus et al., Nat comm 2019).

We have included this study in our Introduction on page 2 lines 98-102. We have also performed additional simulations of the double mutant F309P/F313P where the hydrophobicity of H8 is disrupted. We simulated this mutant in POPC, SOPC, SOPC + 10 mN/m and SOPC:SOPE membranes. We have added a new text describing those systems starting at page 8 line 483-534. We also added 3 additional figures (S5-S7) in the Supplementary Material.

The authors use the AngII-bound 6OS0 PDB structure as a canonical active state, but other structures with different bias agonist profiles exist (6OS1 and 6OS2). I wonder how the active conformation(s) stabilized by PE and tension would compare to those structures.

Despite having different agonists bound, these three structures are remarkably similar with RMSD values of 0.82 Å for 6OS1 relative to 6OS0 and 0.74 Å for 6OS2. Therefore, we don't expect significant differences for using any of these 3 structures as a reference for our measurements. Wingler et al. (Science 2020) discuss this issue in their work as one would expect that the receptor would have different structural features when bound to different agonists. However, binding of the nearly-identical nanobodies likely limits the structural conformation that the receptor can adopt in the crystal structures.

Reviewer 3

The manuscript presented by Poudel et al describes different activation states of the AT1 receptor under different membrane compositions and the addition of membrane tension. The study is solely based on molecular dynamics simulations which uses hallmark structural changes taken from the available crystal structures of the active and inactive state to evaluate the different membrane effects.

Key findings include the fact that membrane thickness is a key player in the activation state of the AT1R, which should serve as an explanation for mechano-stimulation of this receptor. This notion would for sure be of high interest if it would be supported by experimental data. But as it is the paper yields more questions than answers. The fact itself that mechanical stress induces distinct conformational changes in AT1R has already been shown through MD simulations by Yasuda et al in 2008 (DOI: 10.1038/sj.embor.7401157), a study that was not cited by the authors.

We have now included the suggested paper by Yasuda et al. in the Introduction (page 2 lines 94-98). We want to point out that the Yasuda et al. study did not perform MD simulations of the AT1 receptor as we have done in our current study. Yasuda et al. generated a basic molecular model based on homology modeling and their experimental observations. This model did not even include a membrane or solvent and it was only intended for illustration purposes.

Even though I am sure there is more to describe in this process, the authors should support their models by experimental data. Artificial membrane-like environments could be used for such a study.

We have made our best effort to present our modelling work within the context of experiments on the AT1 receptor as well as other important GPCRs such as rhodopsin. Our group does purely computational research and therefore we do not have the resources or access to facilities to perform the suggested experiments. We are exploring collaborations with experimental groups for future publications.

A more general question is where the applied settings for the simulation actually play a physiological role. Are the used short and long fatty acids representative for a certain tissue or cell type?

Is it feasible to assume that pure changes in lipid thickness can appear that induce a change distinguishable from other influencing factors such as headgroup charges or protein and cholesterol distribution?

We have addressed both of these comments in the Discussion section (page 12, lines 692-712)

How can membrane tension lead to a thinner membrane when the same fatty acids are present? Is there a change in the arrangement?

The lipid membrane is largely incompressible and therefore the volume must be conserved during a deformation. Therefore, as tension is applied and the area of the membrane increases, there must be a corresponding change in the thickness of the membrane (see page 7, lines 409-414).

How do the enormously high-tension ranges of 5 to 20 mN/m that are required for the simulation relate to physiological force ranges?

From an experimental perspective it is currently unknown what is the tension needed to modulate the activation of the AT1 receptor in model systems as studies have not determined pressure-response curves (to the best of our knowledge). In bacterial mechanosensitive channels, the response thresholds may be in the range of 7-15 mN/m. In simulations, however, the values of tension needed to induce a particular response on a protein depend on system size (including the number of lipids and the overall size of the protein). This is because the small simulation sizes limit long-range fluctuations due to the imposed periodic boundary conditions. We have tried to make this more clear on page 6, lines 360-364.

Overall, there are some interesting aspects of the paper that once supported by experimental data might provide a deeper understanding of the membrane-induced effects on AT1R signaling.

REVIEWER COMMENTS

Reviewer #1 (Remarks to the Author):

The authors have improved the manuscript by addressing the reviewer's comments and analyzing the additional simulations. A number of remaining comments include:

1. Deactivation of even agonist-bound class A GPCRs in POPC lipids upon removal of G protein have been observed in previous longer MD simulations beyond 1000ns using the Anton supercomputer or enhanced sampling simulations within 1000ns. It's unclear whether the conclusions of this study can be affected by the lengths of the simulations, especially for apo AT1 in Fig. 1 and agonist-bound AT1 in Fig. 4.

Relevant references include:

Dror RO, Arlow DH, Maragakis P, Mildorf TJ, Pan AC, Xu H, Borhani DW, & Shaw DE (2011) Activation mechanism of the β 2-adrenergic receptor. *Proc. Natl. Acad. Sci. U.S.A.*, 108(46):18684-18689.

Miao Y & McCammon JA (2016) Graded activation and free energy landscapes of a muscarinic G-protein-coupled receptor. *Proc Natl Acad Sci U S A*, 113(43):12162–12167.

2. Given the additional simulations, it is important and helps to plot their time courses in at least Supporting Material as in the main Figures 1, 4-5. Do they give similar results for each system?

3. In Fig. 1f, the box and whiskers plots are missing for the blue and red simulations. Seems no data were plotted for some time windows of the simulations; or is the corresponding ICL2 helicity zero?

4. In Figs. S2 and S6, the two dash lines need to be explained. Do the plots of two different colors in Fig. S2 correspond to two replica simulations?

Reviewer #2 (Remarks to the Author):

The authors have satisfactorily addressed my main concern regarding increasing MD simulations times and included 1 replica for each system. Interpretation of trajectories is better supported now. The authors went beyond by analyzing Helix 8 mutations showing a possible force-sensing mechanism involving the interaction of this helix with lipids' head-groups.

One remaining concern is whether the different structural states captured by MD do indeed correspond to genuinely active states, as the computational approaches were not directly aimed at testing the ability of these states to interact with downstream G proteins. The authors' choice of words in the discussion L743, "active-like states", nicely recapitulates this uncertainty. I would therefore suggest the authors to change their title to:

"Membrane mediated mechanical stimuli produces distinct active-like states in the AT1 receptor".

Reviewer #3 (Remarks to the Author):

The authors address several issues raised from the reviewers with respect to the modelling. While I appreciate the reference to experimental data on e.g. rhodopsin, I consider the complete lack of experimental data to evaluate the hypotheses from the computational model a severe drawback of the study. Science is interdisciplinary and collaborations with wet lab groups is not out of the scope for such a paper. Simply stating that this will be explored for future studies is not sufficient.

Any changes in the text are highlighted in teal color to make them easier to identify. We address the reviewer's comments specifically below.

Reviewer 1

The authors have improved the manuscript by addressing the reviewer's comments and analyzing the additional simulations. A number of remaining comments include:

1. Deactivation of even agonist-bound class A GPCRs in POPC lipids upon removal of G protein have been observed in previous longer MD simulations beyond 1000ns using the Anton supercomputer or enhanced sampling simulations within 1000ns. It's unclear whether the conclusions of this study can be affected by the lengths of the simulations, especially for apo AT1 in Fig. 1 and agonist-bound AT1 in Fig. 4.

Relevant references include:

Dror RO, Arlow DH, Maragakis P, Mildorf TJ, Pan AC, Xu H, Borhani DW, & Shaw DE (2011) Activation mechanism of the β 2-adrenergic receptor. *Proc. Natl. Acad. Sci. U.S.A.*, 108(46):18684-18689.

Miao Y & McCammon JA (2016) Graded activation and free energy landscapes of a muscarinic G-protein-coupled receptor. *Proc Natl Acad Sci U S A*, 113(43):12162–12167.

We have incorporated these references in our Results section. We have also performed 3 additional simulations using the Anton 2 supercomputer to address the long-term stability of the active and inactive states in the simulations. We ran three 20 microsecond simulations of the AT1 receptor: 1) in SOPC bound to AngII, 2) in SOPC in its apo state, and 3) in SOPC with 10 mN/m tension in its apo state. Our new results show that the active structure is stable for the AT1 receptor bound to AngII as well as in the case of a tensioned membrane in its apo form. We have included a detailed discussion in section 3.5 (pages 12-14), which includes two new figures (Fig. 8 and 9) and one supplemental figure (S14).

2. Given the additional simulations, it is important and helps to plot their time courses in at least Supporting Material as in the main Figures 1, 4-5. Do they give similar results for each system?

We have included in the supplemental material the time courses for the second replicas of Figures 1, 4, and 5. These are Figures S1, S6, and S11 respectively. While there is variability between the time courses for both replicas, they follow similar trends.

3. In Fig. 1f, the box and whiskers plots are missing for the blue and red simulations. Seems no data were plotted for some time windows of the simulations; or is the corresponding ICL2 helicity zero?

The corresponding ICL2 helicity is zero for the blue and red data in that plot.

4. In Figs. S2 and S6, the two dash lines need to be explained. Do the plots of two different colors in Fig. S2 correspond to two replica simulations?

Yes, the two lines represent the data for each replica. We have changed the colors in the plot and clarified this in the caption of now Figures S3 and S9.

Reviewer 2

The authors have satisfactorily addressed my main concern regarding increasing MD simulations times and included 1 replica for each system. Interpretation of trajectories is better supported now. The authors went beyond by analyzing Helix 8 mutations showing a possible force-sensing mechanism involving the interaction of this helix with lipids' head-groups.

One remaining concern is whether the different structural states captured by MD do indeed correspond to genuinely active states, as the computational approaches were not directly aimed at testing the ability of these states to interact with downstream G proteins. The authors' choice of words in the discussion L743, "active-like states", nicely recapitulates this uncertainty. I would therefore suggest the authors to change their title to:

"Membrane mediated mechanical stimuli produces distinct active-like states in the AT1 receptor".

We have adjusted our title accordingly.